# Association of FANCM Mutations with Familial and Early-Onset Breast Cancer Risk in a South American Population

**DOI:** 10.3390/ijms24044041

**Published:** 2023-02-17

**Authors:** Sebastian Morales-Pison, Sarai Morales-González, Ricardo Fernandez-Ramires, Julio C. Tapia, Edio Maldonado, Gloria M. Calaf, Lilian Jara

**Affiliations:** 1Centro de Oncología de Precisión (COP), Universidad Mayor, Santiago 7560908, Chile; 2Facultad de Medicina y Ciencias de la Salud, Universidad Mayor, Santiago 7560908, Chile; 3Laboratorio de Genética Humana, Programa de Genética Humana, Instituto de Ciencias Biomédicas (ICBM), Facultad de Medicina, Universidad de Chile, Santiago 783090, Chile; 4Laboratorio de Transformación Celular, Programa de Biología Celular y Molecular, Instituto de Ciencias Biomédicas (ICBM), Facultad de Medicina, Universidad de Chile, Santiago 783090, Chile; 5Programa de Biología Celular y Molecular, Instituto de Ciencias Biomédicas, Facultad de Medicina, Universidad de Chile, Santiago 783090, Chile; 6Instituto de Alta Investigación, Universidad de Tarapacá, Arica 1010069, Chile

**Keywords:** association study, early-onset breast cancer, FANCM, South American

## Abstract

Breast cancer (BC) is the most common cancer among women worldwide. *BRCA1/2* are responsible for 16–20% of the risk for hereditary BC. Other susceptibility genes have been identified; Fanconi Anemia Complementation Group M (*FANCM*) being one of these. Two variants in *FANCM,* rs144567652 and rs147021911, are associated with BC risk. These variants have been described in Finland, Italy, France, Spain, Germany, Australia, the United States, Sweden, Finnish, and the Netherlands, but not in the South American populations. Our study evaluated the association of the SNPs rs144567652 and rs147021911 with BC risk in non-carriers of *BRCA1/2* mutations from a South American population. The SNPs were genotyped in 492 *BRCA1/2*-negative BC cases and 673 controls. Our data do not support an association between *FANCM* rs147021911 and rs144567652 SNPs and BC risk. Nevertheless, two BC cases, one with a family history of BC and the other with sporadic early-onset BC, were C/T heterozygotes for rs144567652. In conclusion, this is the first study related contribution of *FANCM* mutations and BC risk in a South American population. Nevertheless, more studies are necessary to evaluate if rs144567652 could be responsible for familial BC in *BRCA1/2*-negatives and for early-onset non-familial BC in Chilean BC cases.

## 1. Introduction

Breast cancer (BC) is the most common cancer among women worldwide [1]. A family history of BC is the most important risk factor [2]. It is estimated that genetic predisposition accounts for 5–15% of all BC cases [3,4] and approximately 25% of diagnoses in women under the age of 30 [5]. BRCA1 and BRCA2 are the major high-risk genes that predispose individuals to BC and ovarian cancer (OC). Nevertheless, up to now, mutations in *BRCA1/2* genes are responsible for only 16–20% of the risk for hereditary BC on average [6,7,8]. In recent years, other susceptibility genes with a more moderate effect have been identified to predispose to hereditary BC and OC. Most of these genes are coding for tumor suppressors that function in genome maintenance by promoting homologous recombination repair after DNA double-strand breaks [9]. These genes included *RAD51C*, *RAD51D*, *CHEK2*, *PALB2*, *ATM*, *BRIP1*, *BARD1*, *TP53*, *CDH1*, *RECQL*, and *NBN*, among others [10]. Therefore, in approximately 60% of hereditary BC patients, the genetic predisposition remains unknown.

Fanconi Anemia (FA) Complementation Group M (*FANCM*) has recently been reported as one such susceptibility gene [11]. *FANCM* participates in the DNA damage and repair pathway, being responsible for the repair of the DNA inter-strand crosslinks through homologous recombination and recently has been identified as a tumor suppressor gene [12,13]. *FANCM* deleterious variants are associated with BC susceptibility in monoallelic mutation carriers [9] and particularly with the risk of triple-negative BC subtype [10]. Two truncating germline variants in *FANCM* are associated with BC risk. These mutations correspond to rs144567652 (c.5791C>T) and rs147021911 (c.5101C>T). *FANCM* rs144567652 (pArg1931*) is a nonsense mutation that causes the skipping of exon 22, which generates a protein that lacks DNA repair activity [14]. This variant is a familial BC risk factor and has been described in Finland, Italy, France, Spain, Germany, Australia, the United States, Sweden, and the Netherlands [14]. The rs147021911 (p.Gln1701*) mutation, located in exon 20, was identified in *BRCA1/2*-negative BC patients. Subsequently, a genotyping study carried out in a large series of Finnish BC patients, and healthy controls showed a significant association (OR = 1.86 [95% CI = 1.26–2.75, *p* = 0.0018]) with disease risk, especially for triple-negative BC [15]. The rs144567652 and rs147021911 *FANCM* mutations have not yet been described in the Asian, African, or South American populations. The Chilean population is the result of admixture between the Asian and Spanish populations; therefore, whether germline *FANCM* mutations might contribute to BC risk in Chilean individuals is unknown. The aim of this study was to determine the contribution of these two *FANCM* variants to familial BC and early-onset non-familial BC in Chilean non-carriers of *BRCA1/2* mutations.

## 2. Results

The observed genotype frequencies for the two SNPs were in Hardy-Weinberg equilibrium in controls (*p* = 1.0 for rs147021911 and *p* = 0.98 for rs144567652). For the case-control analysis, the whole BC sample was subdivided into two subgroups: individuals from families with two or more members with BC and/or OC (*n* = 314) (subgroup A) and individuals with non-familial early-onset BC (≤50 years) (*n* = 178) (subgroup B). Subgroup A excludes the subgroup B cases.

The single-locus analysis indicated that only the wild-type allele (allele C) of rs147021911 was present in both cases and controls in this Chilean population. The frequency of allele C was 1.0 in both cases and controls, and 100% of BC patients in subgroups A and B were C/C homozygotes. Therefore, no association between *FANCM* rs147021911 and BC risk was found in the sample. Table 1 shows the allele and genotype distributions of the *FANCM* rs144567652 polymorphism. As shown, there were no significant differences between cases and controls, either in the whole group or subgroup analyses (*p* > 0.05). A total of 99.6% of the total cases and 99.9% of controls were homozygous for the wild-type allele (C/C). Among the cases, two patients were C/T heterozygotes, corresponding to 0.4% of total cases, and only one heterozygote was detected among the controls (0.1%). No T/T homozygotes were detected among cases or controls.

We also analyzed the relationship between the rs144567652 and BC risk within cases with BC family history according to the number of BC cases in the family (Table 2). In the families with 2 BC and/or OC cases, 98.8% of the cases and 99.9% of controls were homozygous for the wild-type allele (C/C), and 100% of the BC cases were C/C in the families with ≥3 BC and/or OC cases. Therefore, no association was found between this SNP and BC risk in cases with moderate (2 BC and OC cases) or strong (≥3 BC and OC) family history. Interestingly, the two heterozygous cases C/T belong to the same group of cases with a moderate family history of BC.

Figure 1 shows the frequencies of rs144567652 (c.5791C>T) in the countries where association studies have been carried out between this SNP and BC risk [10,14,16]. Studies have only been carried out in some European countries such as Spain, France, Italy, Germany, Netherlands, Finland, and Sweden, and only in the United States in North America and Australia. No studies of this SNP have been carried out in other continents like Asia, Africa, and Central and South America.

In individuals of the general population, the frequency of rs144567652 mutation carriers is only described in Europe countries such as Finland (0.24), Germany (0.05), France (0.08), and Italy (0.07). There are no data regarding the Spanish population. In Australia, the frequency was 0.09. In Chile, our study shows that the frequency of rs144567652 in controls was 0.14 (Figure 1). In European cases with BC, carriers of the mutation rs144567652 were detected in Finland, Spain, France, and Italy, with an average frequency of 0.39. In the United States, no cases of BC carriers of the mutation rs144567652 were detected. In this study, we observed that de frequency in BC Chilean cases was 0.40, which is the same frequency described in Spain BC cases. In Sweden and the United States, the mutation was not present in BC cases or in the general population (Figure 1). Then, this is the first association study of the SNP rs144567652 in a South American population.

The two BC cases heterozygous for *FANCM* rs144567652 were negative for pathogenic *BRCA1/2* mutations, confirmed by Next Generation Sequencing (NGS) (Axen BRCA Panel, Macrogen, Seoul, South Korea). Figure 2A displays the pedigrees of case F188p1 and the Sanger sequencing confirmation showing the presence of the T allele of the SNP, and Figure 2B displays the pedigrees and Sanger sequencing confirmation for case F579p1. Proband F188P1 is a patient belonging to a family with BC (3 cases), lung cancer (1 case), testicular cancer (2 cases with very early diagnosis), colon cancer (one case), and melanoma (one case); this patient was diagnosed with early onset of BC (34 years old) and positive for estrogen and progesterone receptors with negative lymph nodes. Proband 579P1 also had a family history of cancer, including BC (2 cases), prostate cancer (2 cases), oral cancer (one case), and colon cancer (one case); this patient presented a molecular profiling of positive for estrogen and progesterone receptor with bilateral infiltrating ductal carcinoma and with negative lymph nodes, diagnosed at 56 years of age.

The heterozygous control was *BRCA1/2*-negative for pathogenic mutations, confirmed by NGS (Axen BRCA Panel, Macrogen, Seoul, South Korea). Her family reported no cases of BC, OC, or prostate cancer. There was one uterine cancer case in a maternal aunt, diagnosed at 68 years old.

## 3. Discussion

Currently, there is consensus that *BRCA1* and *BRCA2* mutations account for approximately 16% of the risk for familial BC and OC [6,7,8]. Consequently, there is an intensive search for additional susceptibility targets. A study that screened *BRCA1/2*-negative women using a combination of multi-gene panel sequencing and comparative genomic hybridization showed a remarkably high frequency of a truncating variant in *FANCM* [17]. Further, Nguyen-Dumont et al. (2018) searched for evidence using case-control analyses screening among 427 women with BC of Polish and Ukrainian population identifying one carrier of the *FANCM* nonsense mutation c.1972C>T with a frequency of 0.23% and two carriers of the frameshift insertion c.1491dup with a frequency of 0.47% [18].

*FANCM* is part of the FA complementation group along with 21 more *FANC* genes described to date [19,20,21,22,23]. Mutations in these groups of genes are associated with FA development which is characterized by an early onset of aging, severe bone marrow failure, and an extremely high predisposition to various cancers [24].

*FANCM* is composed of 23 exons that encode for five domains, the DEAH helicase domain, which is important for the FA complex to monoubiquitinate FANCI-FANCD2, MM1-3 motifs and the ERCC4-like endonuclease domain (Figure 3). *FANCM* has recently been suggested as a susceptibility gene and as a tumor suppressor gene for hereditary BC [17].

Three *FANCM* nonsense mutations, rs147021911 (c.5101C>T), rs144567652 (c.5791C>T), and c.4025_4026delCT, were recently found in a Finnish cohort of BC cases [11,15]. Of these variants, the rs147021911 is the most common, with a frequency of 2.8% in unselected patients, 3.1% in BC families, and 5.6% in patients with triple-negative BC [15]. Further studies in other populations have to establish that the two truncating germline variants within the *FANCM* gene, rs147021911 (c.5101C>T) and rs144567652 (c.5791C>T), are associated with increased BC risk [14,15,16].

The *FANCM* rs147021911 nonsense mutation is located in exon 20 within the ERCC4 domain (Figure 3). This mutation was identified by Kiiski et al. (2014) [15] through exome sequencing of germline DNA samples from 24 *BRCA1/2*-negative BC patients. It was further genotyped in a large series of Finnish BC patients and healthy controls [15]. The mutation was found to be associated with BC [OR = 1.86, 95% CI = 1.26–2.75, *p* = 0.0018], especially among triple-negative BC cases with positive estrogen and progesterone receptors and negative for HER2 (OR = 3.56, 95% CI = 1.81–6.98, *p* = 0.0002). Further, this mutation was found in 3.39% of BRCA1/2-negative familial BC cases (OR = 2.11, 95% CI = 1.34–3.32, *p* = 0.001) with an even higher carrier frequency of 5.88% (12 in 204) observed in a subgroup of mainly unselected cases with a triple-negative BC tumor phenotype (OR = 3.56, 95% CI = 1.81–6.98, *p* = 0.01) [15]. In 2016, Kiiski et al. studied this mutation in association with the disease prognosis [25]. The authors show that the rs147021911 was associated with poor 10-year BC survival (HR = 1.66, 95% CI = 1.09–2.52, *p* = 0.018] and an especially reduced survival in patients who had not received radiotherapy (HR = 3.43, 95% CI = 1.6–7.34, *p* = 1.5 × 10^−3^), suggesting that rs147021911 carriers have a reduced BC survival, but postoperative radiotherapy may diminish this survival disadvantage [25]. In this study, we genotyped *FANCM* rs147021911 in 492 *BRCA1/2*-negative BC cases and 673 controls. One hundred percent of the cases and controls were C/C. Considering the small sample set and the fact that this is a rare variant in the majority of the populations, this result could be expected. According to the Ensembl database, this variant is monomorphic in controls from the five super populations in the 1000 Genomes Project (Africans, Admixed Americans, East Asians, Europeans, and South Asians). With respect to BC cases, according to the ExAC database [26], this mutation is more common in Finland (carrier frequency ~1.8%) than in other European populations (carrier frequency ~0.3%). The contemporary Chilean population was produced by an admixture of Amerindian peoples with sixteenth- and seventeenth-century Spanish settlers. Later (nineteenth-century) immigration from Germany, Italy, Croatia, and Middle Eastern nations had a negligible effect on the ethnic makeup of the country (representing less than 4% of the national population), and any impact was largely circumscribed to the localities where the immigrants were concentrated [27,28,29]. The modern Chilean population is ~52% Caucasian and ~44% Native American [27,28,29]. It is likely that this mutation is either absent from the Chilean population or present at an incidental frequency, and therefore it is not a probable risk factor for BC. This is the first case-control study between *FANCM* rs147021911 mutations and BC in South America.

The *FANCM* variant rs144567652 generates a premature stop codon, resulting in the loss of 118 amino acids from the C-terminal. The functional characterization of the mRNA transcript derived from the rs144567652 allele shows that the variant causes the skipping of exon 22, introducing a premature stop codon. In addition, genetic complementation assays revealed the mutated protein lacks DNA repair activity. These findings support the idea that the *FANCM* rs144567652 mutation is pathogenic. Gracia-Aznarez et al. (2013) [16] studied the variant rs144567652 in 3409 *BRCA1/2*-negative familial BC cases and 3896 controls from Italy, the Netherlands, Australia, and Spain, and the variant was detected in 10 cases (0.3%) and 5 controls (0.1%), with an estimated OR of 2.29 (95% CI = 0.71–8.54, *p* = 0.13). These authors detected a low frequency of this variant (0.0011 in cases and 0.00077 in controls), with a variable prevalence in the available populations (0.6% in the Italian and 0.3% in the Spanish samples, not found in the Dutch sample and present in only one Australian control (0.14%). Given the important role of *FANCM* in DNA repair, Gracia-Aznarez et al. (2013) suggested that it would be interesting to analyze this gene in a greater number of samples to better understand its contribution to BC [16]. Peterlongo et al. (2015) [14] performed a multinational study genotyping the rs144567652 variant in 8635 *BRCA1/2*-negative familial BC cases and 6625 controls from Italy, France, Spain, Germany, Australia, the United States, Sweden, and the Netherlands, confirming an association between truncating *FANCM* mutations and BC risk, with a carrier frequency of 0.21% in cases and 0.06% in controls, and an OR of 3.93 (95% CI = 1.28–12.11; *p* = 0.017). When the Peterlongo et al. data were combined with those of Gracia-Aznarez et al., overall, the variant was detected in 28 of 12,044 (0.23%) familial cases and 9 of 10,521 (0.09%) controls, corresponding to an OR of 2.83 (95% CI = 1.33–6.01, *p* = 0.007) [14,16]. It must be noted that these studies were based on cases with a positive family history of BC and/or early-onset disease. These selected cases are likely to be enriched in predisposing genetic factors. Consequently, the ORs observed here could be higher than those expected in unselected case and control populations. In the present study, the variant rs144567652 was detected in 2 of 492 (0.4%) and 1 of 673 (0.1%) controls; therefore, these frequencies are higher than those reported in European populations. There are no data regarding this mutation in the general Spanish population, and the frequency reported in BC cases was 0.40. In the Chilean population, the frequency of the rs144567652 mutation was 0.40, equal to the frequency described for Spanish BC cases. Interestingly, this mutation was not detected in patients with early-onset sporadic BC, and the two cases belonging to different families were carriers of the rs144567652 variant, both of whom were negative for *BRCA1/2* mutations. Proband F188P1 was an early-onset familial BC case, and F579P1 presented with bilateral BC. Ultimately, no significant association was found between the rs144567652 mutation and familial BC risk, likely due to the sample sizes used. Therefore, more studies are necessary to evaluate if rs144567652 could be responsible for familial BC in *BRCA1/2*-negatives and for early-onset non-familial BC in Chilean BC cases.

## 4. Materials and Methods

### 4.1. Families

A sample of 492 *BRCA1/2*-negative BC patients from Chilean families was enrolled from Corporación Nacional del Cancer (CONAC) files. None of the families fulfilled the criteria for other BC-related syndromes, including ataxia-telangiectasia, Li-Fraumeni, or Cowden syndrome. Table 3 shows the characteristics of the selected families according to the inclusion criteria. All families included in this study have self-reported Chilean ancestry dating from several generations, validated through extensive interviews with several members of each family. In the selected families, 63.4% have cases of both BC and ovarian cancer (OC). In the BC group, the mean age at diagnosis was 42.8 years of age, and 36.6% had an age of onset < 50 years. The study was approved by the Institutional Review Board of the University of Chile School of Medicine (Project 1200049, 1 March 2020). Informed consent was obtained from all participants.

### 4.2. Controls

A healthy Chilean control sample (*n* = 673) was recruited from CONAC files. DNA samples were taken from unrelated individuals with no personal or familial history of cancer who provided consent for anonymous testing. These individuals were interviewed and informed as to the aims of the study. DNA samples were obtained according to all ethical and legal requirements. The control sample was matched to the case population for age and socioeconomic strata.

### 4.3. FANCM rs147021911 and rs144567652 Genotyping

Genomic DNA was extracted from peripheral blood lymphocytes of 492 BC cases belonging to the selected high-risk families and 673 controls. Samples were obtained as described by Chomczynski and Sacchi [30]. rs147021911 and rs144567652 genotyping were performed using the commercially available TaqMan Genotyping Assay (C_163159164_10 and C_163079670_10, respectively) in a StepOnePlus Real-Time thermal cycler (Applied Biosystems, Foster City, CA, USA) for BC cases and controls. The reaction was performed in a 10-uL final volume containing 1X TaqMan Genotyping Master Mix, 20X TaqMan SNP Genotyping Assay, and 5 ng of genomic DNA. The thermal cycle protocol was as follows: 10 min at 95 °C, followed by 40 cycles each of 92 °C for 15 s and 60 °C for 1 min. Each genotyping run included control DNA confirmed by Sanger sequencing. StepOne version 2.2 was used to allocate the alleles (Applied Biosystems, Foster City, CA, USA). Genotyping was repeated on 10% of the samples as quality control, and all genotype grading was completed and confirmed separately by two reviewers blind to case-control status.

### 4.4. Sanger Sequencing Confirmation

Sanger sequencing was performed to confirm the heterozygous C/T genotypes of rs144567652. First, we designed one pair of primers in the laboratory using Primer3 v.0.4.0 (Whitehead Institute for Biomedical Research, Cambridge, UK; Boston, MA, USA) (forward: 5′-AGTTTGCTCAATGGTGTAGATCT-3′ and reverse: 3′-TGTTAGCCATCCTTTTCACAGAT-5′). A 402-pb PCR fragment containing rs144567652 was generated by standard polymerase chain reaction (PCR). Sanger sequencing was performed in an ABI 3730xl automated fluorescence-based sequencer and BigDye v.3.1 terminator system (Applied Biosystems, Foster City, CA, USA).

### 4.5. Statistical Analysis

The Hardy-Weinberg equilibrium assumption was assessed in the control sample using a goodness-of-fit chi-square test (HW Chisq function included in the ‘Hardy Weinberg’ package v1.4.1 for R, Foundation for Statistical Computing, Vienna, Austria, URL: https://www.r-project.org/ accessed on 29 June 2022). Fisher’s exact test was used to test the association between genotypes/alleles and case/control status. Odds ratios (OR) with 95% confidence intervals (CI) were calculated to estimate the strength of the associations (OR and Fisher’s functions performed using GraphPad Prism v 6.0 for Windows 10 (GraphPad Software, La Jolla, CA, USA). A two-tailed *p*-value < 0.05 was used as the criterion of significance.

## Figures and Tables

**Figure 1 ijms-24-04041-f001:**
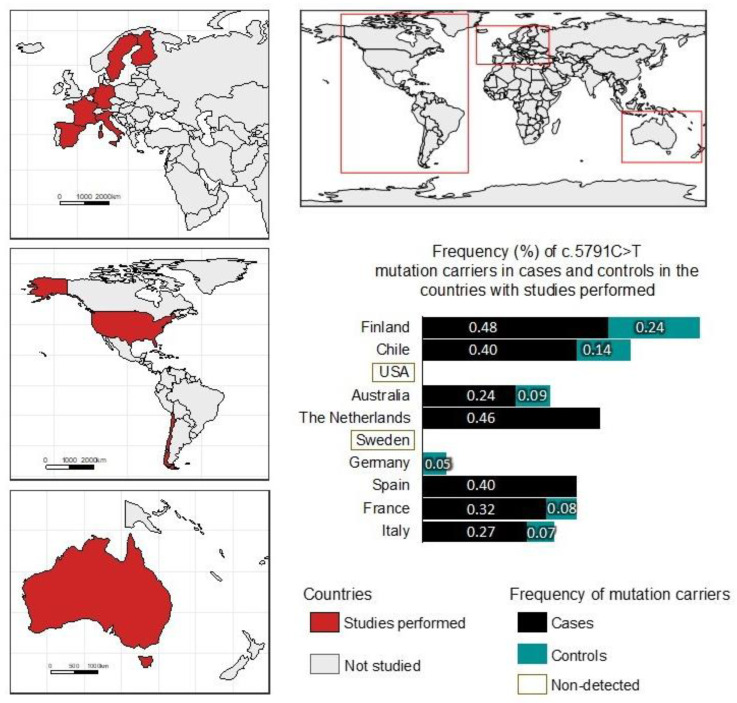
Frequencies (%) of carriers of the rs144567652 (c.5791C>T) variant in breast cancer cases and controls from different countries. The red-filled countries correspond to the countries in which association studies between the SNP rs144567652 and BC risk have been performed. The light-yellow-colored boxes indicate those countries in which the rs144567652 variant was studied but not detected.

**Figure 2 ijms-24-04041-f002:**
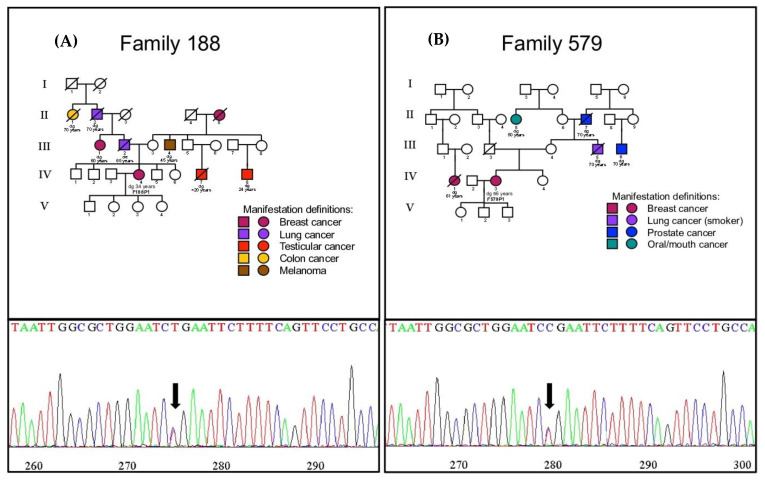
Pedigree and Sanger confirmation of rs144567652 heterozygous cases. (**A**) correspond to the pedigree of the F188 case (upper panel) and her Sanger confirmation (lower panel, arrow show the heterozygous variant). (**B**) correspond to the pedigree of the F579 case (upper panel) and her Sanger confirmation (lower panel, arrow show the heterozygous variant).

**Figure 3 ijms-24-04041-f003:**
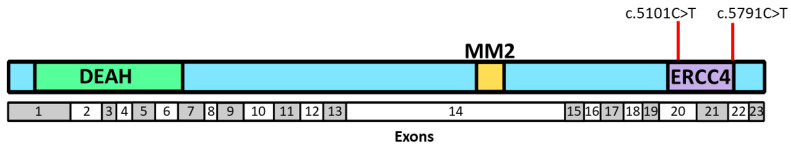
Schematic illustration of FANCM domains and rs147021911 (c.5101C>T) and rs144567652 (c.5791C>T) localization.

**Table 1 ijms-24-04041-t001:** Genotype and allele frequencies of rs144567652 (FANCM) in BRCA1/2-negative breast cancer cases and controls.

		All BC Cases (*n* = 492)			Families with ≥2 BC and/or OC Cases (*n* = 314)			Families with a Single Case, Diagnosis at ≤50 Years of Age (*n* = 178)		
Genotype or Allele	Controls (%) (*n* = 673)	BC Cases (%)	OR [95% CI]	*p*-Value ^a^	BC Cases (%)	OR [95% CI]	*p*-Value ^a^	BC Cases (%)	OR [95% CI]	*p*-Value ^a^
C/C	672 (99.9)	490 (99.4)	(ref)	-	312 (99.4)	(ref)	-	178 (100)	(ref)	-
C/T	1 (0.1)	2 (0.4)	2.7 [0.3–39.8]	0.5	2 (0.6)	4.3 [0.4–62.5]	0.2	0 (0.0)	-	-
T/T	0 (0.0)	0 (0.0)	-	-	0 (0.0)	-	-	0 (0.0)	-	-
C/T + T/T	1 (0.1)	2 (0.4)	2.7 [0.3–38.8]	0.1	2 (0.6)	4.3 [0.4–62.5]	0.2	0 (0.0)	-	-
Allele C	1345 (99.9)	982 (99.8)	(ref)	-	626 (99.7)	(ref)	-	356 (0.0)	(ref)	-
Allele T	1 (0.1)	2 (0.2)	2.7 [0.3–39.7]	0.5	2 (0.3)	4.2 [0.4–62.3]	0.2	0 (0.0)	-	-

BC—breast cancer, OC—ovarian cancer, OR—odds ratio, CI—confidence interval, Ref—reference; ^a^ Fisher’s exact test; (*p* < 0.05).

**Table 2 ijms-24-04041-t002:** Genotype and allele frequencies of rs144567652 (*FANCM*) according to the number of BC cases per family in BRCA1/2-negative breast cancer cases and controls.

		Families with 2 BC and/or OC Cases (*n* = 166)			Families with ≥3 BC and/or OC Cases (*n* = 148)		
Genotype or Allele	Controls (%) (*n* = 673)	BC Cases (%)	OR [95% CI]	*p*-Value ^a^	BC Cases (%)	OR [95% CI]	*p*-Value ^a^
C/C	672 (99.9)	164 (98.8)	(ref)	-	148 (100)	(ref)	-
C/T	1 (0.1)	2 (1.2)	8.1 [0.9–118.9]	0.1	0 (0.0)	-	-
T/T	0 (0.0)	0 (0.0)	-	-	0 (0.0)	-	-
C/T + T/T	1 (0.1)	2 (1.2)	8.1 [0.9–118.9]	0.1	0 (0.0)	-	-
Allele C	1345 (99.9)	330 (99.4)	(ref)	-	296 (100)	(ref)	-
Allele T	1 (0.1)	2 (0.6)	8.1 [0.9–118.2]	0.1	0 (0.0)	-	-

BC—breast cancer, OC—ovarian cancer, OR—odds ratio, CI—confidence interval, Ref—reference; ^a^ Fisher’s exact test; (*p* < 0.05).

**Table 3 ijms-24-04041-t003:** Inclusion criteria for study families.

Inclusion Criteria	Families: *n*
Three or more family members with breast and/or ovarian cancer	148 (29.8%)
Two family members with breast and/or ovarian cancer	166 (33.6%)
Single affected individuals with breast cancer aged ≤35	87 (17.9%)
Single affected individuals with breast cancer aged 36–50	91 (18.7%)
TOTAL	492 (100%)

## Data Availability

All data are shown within the manuscript.

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
