# Peer review of "Association of FANCM Mutations with Familial and Early-Onset Breast Cancer Risk in a South American Population"

_ijms, 2023, doi:10.3390/ijms24044041_

Round 1

Reviewer 1 Report

This paper is the first to study the frequency of two known FANCM variants in a South American population (SA). These two variants were known from earlier studies to be associated with a modest increase in breast cancer risk (OR<2). These variants are rare in all populations, mostly with a population frequency < 1% and with this small sample size it was not possible to confirm this relatively low risk. Allele frequencies in gnomAD for these two are for rs147021911 0.01% and for rs144567652 0.03%. The finding of none with rs147021911 in the 492 cases was expected and to find two with rs144567652 was unexpected and suggest this could be more common. Still, this is the first report of these two variants in this population.

Comments:

There is no need to talk about monomorphic for any of the alleles – these are rare variants in all populations and it is not expected to find them in small sample sets.

Table 2 is not commented on under results? 

I have a problem with figure 1 

Looks odd that Sweden is red but no numbers in the table? Finland is not red although most studies of these two variants were done in Finland. USA is red but with no numbers?

This data could be corrected and presented in a table with country, frequency of mutations and references to where the data was published. But could also be omitted.

End of Discussion is suggested that the variant could be responsible for all cancers in the two families. I think this should be modified. The variant is associated with a slightly increase in risk in breast cancer and is not suggested as causing a high-risk cancer syndrome.

As a whole, Discussion could be shortened. 

Author Response

We thank to the three reviewers for their exhaustive analysis of our manuscript, which has been very useful for us to improve it. We have carefully reviewed the comments made and have responded to all of them.

Reviewer 1

Regarding the observations (OBS) of reviewer 1 we respond in order:

OBS 1: “There is no need to talk about monomorphic for any of the alleles – these are rare variants in all populations and it is not expected to find them in small sample sets.”

Thank you very much for your observation, we have modified the text considering your suggestions made.

OBS 2: “Table 2 is not commented on under results?” 

Thank you very much for your comment, we have added the description of table 2 which was not added properly in the first version.

OBS 3: “I have a problem with figure 1. Looks odd that Sweden is red but no numbers in the table? Finland is not red although most studies of these two variants were done in Finland. USA is red but with no numbers? This data could be corrected and presented in a table with country, frequency of mutations and references to where the data was published. But could also be omitted.”

Thank you very much for your observation, the figure has been corrected and references have been added to improve understanding from where the data was collected to make the figure.

OBS 4: “End of Discussion is suggested that the variant could be responsible for all cancers in the two families. I think this should be modified. The variant is associated with a slightly increase in risk in breast cancer and is not suggested as causing a high-risk cancer syndrome.”

Thank you very much for your suggestion. The discussion was corrected in the final part to better understand the results obtained.

Reviewer 2 Report

In the study entitled: “Association of FANCM mutations with familial and early-onset breast cancer risk in a South American population” the authors have evaluated the association of the SNPs rs144567652 and rs147021911 with BC risk in non-carriers of BRCA1/2 mutations from a South American population. The SNPs were genotyped in 492 BRCA1/2-negative BC cases and 673 controls. The rs147021911 was monomorphic in the Chilean population and did not support an association between FANCM 26 rs144567652 SNP and BC risk. However, the rs144567652 mutation could be responsible for familial BC in BRCA1/2- negatives cases and the cause of the early-onset non-familial BC.

Some obstacles prevent the manuscript from being published in its current form.

These, among others, include:

Introduction

The entire gene or protein name should be written in parenthesis for the FANCM when mentioned for the first time in the manuscript text.

Line 49-51… The authors have written: “The rs147021911 (p.Gln1701*) mutation, located in exon 20, was identified in BRCA1/2-negative BC patients. Subsequently, a  genotyping study carried out in a large series of Finnish BC patients, and healthy controls showed a significant association (OR=1.86 [95% CI=1.26-2.75, p=0.0018]) with disease risk, especially for triple-negative BC.”- The references are missing.

Materials and Methods

The authors have written.: “A sample of 492 BRCA1/2-negative BC patients from Chilean families were enrolled from Corporación Nacional del Cancer (CONAC) files.” Therefore, in tables 1, 2, and 3, the expression and/or could be misleading and should be modified accordingly. 

The catalog number of FANCM rs147021911 and rs144567652 TaqMan SNP Genotyping Assay should be provided.

Statistical analysis

Was any form of correction (e.g., Bonferroni correction or similar) for multiple comparisons (Table 1 and Table 2) applied?

Results

Line 61-63 …. The authors have written: “For the case-control analysis, the whole BC sample was subdivided into two subgroups: individuals from families with two or more members with BC and/or OC (n=314) (subgroup A), and individuals 63 with non-familial early-onset BC (≤50 years) (n=178) (subgroup B).”- The term and/or is misleading and should be modified accordingly. I presume that all patients were diagnosed with BC. If some patients only have OC, the entire manuscript should be revised. Namely, in the materials and method section, the authors have stated: “A sample of 492 BRCA1/2-negative BC patients from Chilean families were enrolled from Corporación Nacional del Cancer (CONAC) files.”

In table 1 and table 2, the superscript (a) for the p-values should be explained in the table legend.

The percentages in Table 1 should be checked [BC cases (%)---490 (43.8)]

Line 91… The authors have written: “Figure 1 shows the frequencies of rs144567652 in the countries where association studies have been carried out between this SNP and BC risk.” - The references are missing.

Table 2 is not described or mentioned in the manuscript text.

A major revision of the manuscript is recommended.

Author Response

We thank to the three reviewers for their exhaustive analysis of our manuscript, which has been very useful for us to improve it. We have carefully reviewed the comments made and have responded to all of them.

Reviewer 2

Regarding the observations (OBS) of reviewer 2 we respond in order:

OBS 1: “The entire gene or protein name should be written in parenthesis for the FANCM when mentioned for the first time in the manuscript text.”

Thank you very much for your observation, the complete name of FANCM was added in the abstract and within the manuscript.

OBS 2: “Line 49-51… The authors have written: “The rs147021911 (p.Gln1701*) mutation, located in exon 20, was identified in BRCA1/2-negative BC patients. Subsequently, a genotyping study carried out in a large series of Finnish BC patients, and healthy controls showed a significant association (OR=1.86 [95% CI=1.26-2.75, p=0.0018]) with disease risk, especially for triple-negative BC.”- The references are missing.”

Thank you very much for your observation, the missing reference was added.

OBS 3: “The authors have written.: “A sample of 492 BRCA1/2-negative BC patients from Chilean families were enrolled from Corporación Nacional del Cancer (CONAC) files.” Therefore, in tables 1, 2, and 3, the expression and/or could be misleading and should be modified accordingly.”

Thank you very much for your observation. However, the and/or expression is correctly used. The sample of 492 BRCA1/2-negative BC patients includes only index cases with breast cancer. However, in the families of these index cases there are other cases of breast cancer and also cases of ovarian cancer. Consequently, when we mention “families” we refer to the index case, plus the cases of breast cancer and/or ovarian cancer that exist in that family.

OBS 4: “The catalog number of FANCM rs147021911 and rs144567652 TaqMan SNP Genotyping Assay should be provided.”

Thank you very much for the information, the data was added in the corresponding section.

OBS 5: “Was any form of correction (e.g., Bonferroni correction or similar) for multiple comparisons (Table 1 and Table 2) applied?”

Thank you very much for your observation. Bonferroni or other types of Family-wise error rate (FWER) corrections for multiple comparisons (such as Sidak, Benjamini & Hochberg and others) or False Discovery Rate (FDR) are used to deal with make a type-I error when multiple hypotheses are tested. In other words, when multiple variables are tested for association in a case-control study. In our case, we assessed association for only FANCM rs144567652 considering that no carriers of the FANCM rs147021911 variant allele were found. Therefore, a FWER correction is not necessary for our study.

OBS 6: “Line 61-63 …. The authors have written: “For the case-control analysis, the whole BC sample was subdivided into two subgroups: individuals from families with two or more members with BC and/or OC (n=314) (subgroup A), and individuals 63 with non-familial early-onset BC (≤50 years) (n=178) (subgroup B).”- The term and/or is misleading and should be modified accordingly. I presume that all patients were diagnosed with BC. If some patients only have OC, the entire manuscript should be revised. Namely, in the materials and method section, the authors have stated: “A sample of 492 BRCA1/2-negative BC patients from Chilean families were enrolled from Corporación Nacional del Cancer (CONAC) files.”

Thank you very much for your observations. This observation was answered above in OBS 3.

OBS 7: “In table 1 and table 2, the superscript (a) for the p-values should be explained in the table legend.”

Thank you very much for your reviews: the (a) superscript was explained in the table legend.

OBS 8: “The percentages in Table 1 should be checked [BC cases (%)---490 (43.8)].”

Thank you very much for your observations, the percentage values ​​in Table 1 have been reviewed and corrected.

OBS 9: “Line 91… The authors have written: “Figure 1 shows the frequencies of rs144567652 in the countries where association studies have been carried out between this SNP and BC risk.” - The references are missing.”

Thank you very much for your observation, the references were added.

OBS 10: “Table 2 is not described or mentioned in the manuscript text”.

Thank you very much for your comment, we have added the description of table 2 which was not added properly in the first version.

Reviewer 3 Report

This study aims to evaluate the contribution of breast cancer susceptibility gene FANCM in familial BC and early onset non-familial BC cases in Chilean non-carriers of BRCA1/2 population. The authors studied two variants of FANCM - rs144567652 and rs147021911. Authors claim that rs144567652 mutation could be the contributing factor for high cancer frequency and might be the cause of early-onset familial BC in the Chilean population whereas the other variant rs147021911 is not a probable risk factor for BC in the Chilean population.

1.     The paper is extremely confusing. In the abstract (lines 30, 31) authors state that rs144567652 mutation is the cause of familial BC in BRCA1/2 negative population and early-onset non-familial BC but in the conclusions (lines 240, 241) authors conclude that rs144567652 mutation is the cause of early-onset familial BC. Also, in the abstract (lines 26, 27) authors clearly state that their data do not support an association between FANCM rs144567652 and BC risk.

2.     The sample size is small therefore any conclusion drawn from this study sounds potentially overstated.

3.     Line 174 – The sentence is incomplete; authors need to mention the name of cited authors in the sentence (ref.13).

Author Response

We thank to the three reviewers for their exhaustive analysis of our manuscript, which has been very useful for us to improve it. We have carefully reviewed the comments made and have responded to all of them.

Reviewer 3

Regarding the observations (OBS) of reviewer 3 we respond in order:

OBS 1: “The paper is extremely confusing. In the abstract (lines 30, 31) authors state that rs144567652 mutation is the cause of familial BC in BRCA1/2 negative population and early-onset non-familial BC but in the conclusions (lines 240, 241) authors conclude that rs144567652 mutation is the cause of early-onset familial BC. Also, in the abstract (lines 26, 27) authors clearly state that their data do not support an association between FANCM rs144567652 and BC risk.”

Thank you very much for your suggestion. We have made changes to the text that we hope will allow a good understanding of the manuscript.

OBS 2: “The sample size is small therefore any conclusion drawn from this study sounds potentially overstated.”

Thank you very much for your suggestion. However, association studies are usually carried out with n samples of this size, the larger the n sample, the greater the statistical power. The n sample used in our study represents a statistical power of 0.970, which is very good to have an approach to what would be the expected frequencies in a population.

OBS 3: “Line 174 – The sentence is incomplete; authors need to mention the name of cited authors in the sentence (ref.13).”

Thank you very much for your observation, we have proceeded to change and add the name of the cited article.

Round 2

Reviewer 2 Report

The authors have successfully revised the manuscript and explained all the questions or comments raised by reviewers. I have no additional comments.